# Utilization of Cassava Wastewater for Low-Cost Production of Prodigiosin via *Serratia marcescens* TNU01 Fermentation and Its Novel Potent α-Glucosidase Inhibitory Effect

**DOI:** 10.3390/molecules26206270

**Published:** 2021-10-16

**Authors:** Lan Thi Tran, Kuaanan Techato, Van Bon Nguyen, San-Lang Wang, Anh Dzung Nguyen, Tu Quy Phan, Manh Dung Doan, Khamphe Phoungthong

**Affiliations:** 1Faculty of Environmental Management, Prince of Songkla University, Songkhla 90112, Thailand; tranlan02@yahoo.com; 2Environmental Assessment and Technology for Hazardous Waste Management Research Center, Faculty of Environmental Management, Prince of Songkla University, Hat Yai, Songkhla 90110, Thailand; kuaanan.t@psu.ac.th; 3Institute of Biotechnology and Environment, Tay Nguyen University, Buon Ma Thuot 630000, Vietnam; nvbon@ttn.edu.vn (V.B.N.); nadzung@ttn.edu.vn (A.D.N.); dmdung@ttn.edu.vn (M.D.D.); 4Life Science Development Center, Tamkang University, New Taipei City 25137, Taiwan; sabulo@mail.tku.edu.tw; 5Department of Chemistry, Tamkang University, New Taipei City 25137, Taiwan; 6Department of Science and Technology, Tay Nguyen University, Buon Ma Thuot 630000, Vietnam; phantuquy@ttn.edu.vn

**Keywords:** cassava wastewater, fermentation, bioreactor, *S. marcescens*, prodigiosin, α-glucosidase inhibitor, anti-diabetes, docking study

## Abstract

The purpose of this study was to reuse cassava wastewater (CW) for scaled-up production, via the fermentation of prodigiosin (PG), and to conduct an evaluation of its bioactivities. PG was produced at the yield of high 6150 mg/L in a 14 L-bioreactor system, when the designed novel medium (7 L), containing CW and supplemented with 0.25% casein, 0.05% MgSO_4_, and 0.1% K_2_HPO_4_, was fermented with *Serratia marcescens* TNU01 at 28 °C in 8 h. The PG produced and purified in this study was assayed for some medical effects and showed moderate antioxidant, high anti-NO (anti-nitric oxide), and potential α-glucosidase inhibitory activities. Notably, PG was first reported as a novel effective α-glucosidase inhibitor with a low IC50 value of 0.0183 µg/mL. The commercial anti-diabetic drug acarbose was tested for comparison and had a lesser effect with a high IC50 value of 328.4 µg/mL, respectively. In a docking study, the cation form of PG (cation-PG) was found to bind to the enzyme α-glucosidase by interacting with two prominent amino acids, ASP568 and PHE601, at the binding site on the target enzyme, creating six linkages and showing a better binding energy score (−14.6 kcal/mol) than acarbose (−10.5 kcal/mol). The results of this work suggest that cassava wastewater can serve as a low-cost raw material for the effective production of PG, a potential antidiabetic drug candidate.

## 1. Introduction

*Manihot esculenta Crantz* (cassava) is a drought-tolerant crop plant that may be efficiently grown in unfavorable conditions even in marginal areas with poor soil quality and unpredictable rainfall (FAO 2013). It is one of the main staple crops in sub-Saharan Africa and a high-value industrial crop in Asia [1]. Vietnam is ranked as the eighth highest country in terms cassava production [2]. Cassava has, for a long time, been used as a valuable source of human food, or as feed for livestock in developing countries, and applied in the production of starch, biofuels, and some other products (medicines, cosmetics, and biopolymers). The cassava processing industry generates a large amount of wastes/residues that are rich in organic components, and thus, may induce significant environmental issues [3], while these cassava-based industrial wastes (CBIWs) may also be good carbon/nitrogen sources for the production of some valuable products via fermentation [3,4]. CBIWs have been extensively investigated for the bioproduction of biofuels, biogases, biosurfactants, organic acids (citric acid, succinic acid, and lactic acid), volatile fatty acids, and some aroma compounds [3,5,6,7]. In this work, we established a fermentation process at the bench scale (14 L-bioreactor system) for the production of a value-added compound, prodigiosin, from these industrial wastes via bacterial fermentation.

Prodigiosin (PG) is a secondary metabolite synthesized by numerous microbes. Of these, *Serratia marcescens* has been the most widely used in PG production [8]. The utility of PG has been demonstrated in a wide range of potential applications in food colorants, medicines, textiles, agronomy, candles, textiles, and solar cells [8,9,10,11,12]. Due to the various beneficial uses of PG, studies on the biosynthesis of this molecule have recently received renewed and dramatically increased interest [8]. This bioactive compound has been widely produced from some commercial broths such as casein, yeast extracts, tryptone yeast, nutrient broth, tryptone soy, glycerol-tryptone, yeast malt, and peptone-glycerol Luria/Bertani broth [8,9,13,14,15,16]. For low-cost PG biosynthesis, various agro-products and non-traditional substrates such as sesame oil, peanut oil, copra seed, coconut oil, sesame seed, corn steep, peanut seed, crude glycerol, mannitol/corn steep, cassava, and mannitol/cassava have been tested as C/N sources for fermentation [17,18,19,20,21].

Regarding cost-effective PG production and environmental issues, we have previously investigated PG production from various fishery organic wastes, including squid pens, shrimp shells/heads, and crab shells, via *S. marcescens* fermentation [22,23,24,25,26]. In this study, we investigated the potential of cassava-based industrial wastes (cassava wastewater) for PG production and reclamation of its novel potential application as anti-diabetic drug, based on in vitro and docking studies.

## 2. Results and Discussion

### 2.1. Reclamation of Cassava Wastewater as a Low-Cost Substrate for Effective Production of Prodogiosin via Microbial Fermentation

Cassava-based industrial wastes, including cassava residues (CR) and cassava wastewater (CW), were supplemented with free protein (0.5% casein), and then fermented with several *S. marcescens* strains retained from our previous studies. The results are summarized in Table 1. Among the four tested *S. marcescens* strains, *S. marcescens* TNU01 produced the highest PG yield in both media, based on CR and CW. Of these, *S. marcescens* TNU01 produced more PG in the medium that was based on CW than in the medium based on CR, with the PG yields of 3981 and 2191 mg/L, respectively. Therefore, *S. marcescens* TNU01 was chosen as the PG–producing strain, and CW was used as the major substrate in further experiments.

Casein has been found to be a suitable free protein for adding to the media in order to enhance the PG yield [22,23,25,26]. Thus, casein was added to the medium at various concentrations (0.125–0.75%) for fermentation by TNU01 to identify its most appropriate concentration. The results (Figure 1a) indicate that the PG was produced at a high yield (≥3930 mg/L) when the medium was supplemented at concentrations of 0.25–0.75%. As regards cost-effective PG production from wastes, casein—at a 0.25% concentration—was chosen as the best alternative in the design of the fermentation medium.

For comparison, to assess the efficacy of PG production from CW, various commercial materials, including Nutrient broth (NB), Luria-Bertani broth (LB), King’s B, soybean casein digest medium (SCDM), and casein, were also fermented, in the same conditions, with *S. marcescens* TNU01. As shown in Figure 1b, the red pigment was produced at the highest yield in the media that was based on CW and SCDM, with the concentrations of 4012 mg/L and 3987 mg/L, respectively; these PG yields are much higher than those achieved with the other commercial medium alternatives tested (1870–2560 mg/L). The experimental results demonstrate the utility of CW as a low-cost substrate with potential for PG production via microbial technology. 

In our earlier work [24], the bacterium *S*. *marcescens* TNU01 was utilized for the conversion of some organic wastes, including squid pens, shrimp heads, crab shells, and shrimp shells, into PG, with the yields of 2450, 500, 610, and 800 mg/L, respectively. The PG yield was significantly enhanced (to 4015 mg/L) when shrimp shells were supplemented with casein at the mass ratio of 7/3 (shrimp shells/casein) [22]. In this study, we investigated cassava wastewater supplemented with 0.25% casein as a novel C/N source for the cost-effective production of PG, reaching a yield of 4012 mg/L.

### 2.2. The Effects of Supplementary Salts in Culture Medium on PG Production 

Several previous studies have indicated that a suitable salt composition significantly enhances the PG yield from *S. marcescens* fermentation. Supplementary phosphate and sulfate salts have vital roles, via fermentation, in the productivity of PG, but different *S. marcescens* strains or alternative organic wastes, as the major C/N sources for fermentation, may require different kinds of phosphate and sulfate salts at different concentrations [8,22,23,24,25,26]. To explore the choice of suitable salts and their appropriate concentrations in the culture medium, various phosphate and sulfate salts were tested. As shown in Figure 2a, MgSO_4_ was the most suitable source of sulfate in the culture medium, and the optimal concentration was 0.05% (Figure 2b). This sulfate salt (0.05% MgSO_4_) was used in combination with several phosphate salts to screen for a suitable phosphate salt choice for PG biosynthesis. 

The experimental data presented in Figure 2c indicate that the culture medium supplemented with K_2_HPO_4_ gave the highest PG yield production, so K_2_HPO_4_ was chosen as the most suitable phosphate salt for PG production, and its optimal concentration in the culture medium was 0.1% (Figure 2d). Overall, 0.05% MgSO_4_ combined with 0.1% K_2_HPO_4_ was considered to be the most suitable salt combination in the culture medium containing CW as the main substrate; this combination was able to enhance the PG productivity to 5202 mg/L. Even though phosphate and sulfate salts have been widely demonstrated to significantly enhance the PG productivity of *S. marcescens* fermentation in various studies, the mechanisms of this positive effect remain unclear [26].

### 2.3. Scaling-up of PG Production to a 14 L-Bioreactor System and Purification of PG

In fermentation technology, bioreactor systems have been used as strong tools for the scaling-up of bioactive compound production and the reduction in fermentation times [26]. In this study, a 14 L-bioreactor system was utilized for the fermentation of the optimal medium compositions obtained from the experiments above (Section 2.1 and Section 2.2) with *S. marcescens* TNU01. As shown in Figure 3, the PG yield was produced within just 2 h of fermentation and this metabolite was quickly increased at 4 and 6 h of cultivation and reached the highest yield (6150 mg/L) within 8 h of cultivation, with no further PG yield thereafter. Compared to fermentation in flakes (100 mL), the utilization of a 14 L-bioreactor system for the fermentation of medium based on cassava wastewater with *S. marcescens* TNU01 resulted in the production of PG with a significantly higher yield at the larger scale, and with a much shorter fermentation time.

Since numerous potential applications of PG have been demonstrated [8]—especially its effective anticancer activity against various cancers [8,25,27]—without toxicity against normal cells [24], research on the biosynthesis of PG has received renewed interest that has notably increased in recent years [16]. However, in the previous studies, media based on commercial nutrients were widely used as C/N sources for small-scale fermentation in Erlenmeyer flasks to produce bioactive PG [8,9,13,14,15,16]. In contrast, we reused cassava wastewater as the major C/N source for conversion into PG via fermentation, and we scaled-up the production of this pigment compound in a bioreactor system. Some previous studies [22,23,24,26,27,28,29,30,31] also approached the scaling-up of PG biosynthesis in different bioreactor systems (Table 2). Almost all of the studies reported the production of PG in bioreactor systems with true working culture medium volumes of less than 7 L, and with productivity in the range 538–6310 mg/L and fermentation time in the range of 8–65h, except for one previous report [31] that successfully approached the biosynthesis of PG at a 50 L pilot scale, the largest of the reactors used. However, the PG productivity recorded was a low 522 mg/L and the fermentation time was 20 h [31]. In this current study, PG was produced at a large scale (7 L culture medium in a 14 L-bioreactor system) with a high yield (6150 mg/L) in a short cultivation time (8h).

The red pigment compound, produced via fermentation with *S. marcescens* TNU01 in a 14 L-bioreactor system, was extracted and purified according to the methods described in our earlier work [26]. In brief, the fermented culture broth, rich in red pigment content (Figure 4a), was primarily extracted via liquid layer separation (Figure 4b), and the red pigment in the ethyl acetate layer was dried by evaporation to obtain crude red pigment as powder. This crude PG was further purified in an open silica column (Figure 4c) to obtain the purified PG compound. This purified PG was qualified for its purity by running its HPLC profile. As shown in Figure 5a, this purified PG appeared as the major single peak at the retention time (RT) of 12.368 min. This RT of purified PG in this work is approximately similar to those of PG variants obtained in previous studies, showing RT of 12.283, 12.373, and 12.400 min [22,26]. For a more careful confirmation, the PG compound obtained from our previous work was also subjected to HPLC and analyzed at the same conditions, and this reference PG also appeared as a single peak at a similar RT, namely at 12.425 min (Figure 5b). Thus, the PG produced in this study was found to be adequate for biological assays in the subsequent experiments.

### 2.4. Evaluation of Biological Activities of Purified Prodigiosin

The secondary metabolite of PG has been reported for its potential in numerous beneficial applications in various fields, such as in food colorants, medicines, textiles, agronomy, candles, textiles, and solar cells [8,9,10,11,12]. In this work, we reconfirmed and investigated the bioactivities for potential applications of PG in medicines, including anti-NO (anti-nitric oxide), anti-oxidant, and anti-enzyme effects. 

The anti-NO effect has been recognized as an indicator of pro-inflammatory properties, which are relevant to various disorders, including chronic hepatitis, rheumatoid arthritis, and pulmonary fibrosis [32,33]. To assess the anti-NO activity of PG, LPS-stimulated-RAW264.7 cells were tested, and a commercial anti-NO compound, homogentisic acid (HGA), was also used for comparison. As shown in Table 3, both PG and HGA are active anti-NO agents with more than 90% inhibition at the tested concentration of 80 µg/mL. The anti-NO effects of these two compounds were also expressed as IC50 values. PG had comparable anti-NO activity to HGA, with approximately equal IC50 values of 18.06 and 16.12 µg/mL, respectively. PG was first reported as showing potent anti-NO effects with a low IC50 value of 19.1 µg/mL in a prior report [22], and based on the literature review, the anti-NO properties of PG were mentioned in two prior studies [22,26]. The results of this experiment confirm PG as an active anti-NO molecule and also enrich the available data on the anti-NO effects of PG. 

It was evidenced that some major and important kinds of molecules, including the proteins, lipids, and deoxyribonucleic acids of cells, are protected against damage from free radicals by various antioxidant compounds [34]. In this study, antioxidant effects were evaluated via DPPH and ABTS assays, and α-tocopherol, a commercial antioxidant compound, was used for comparison. PG demonstrated great maximum inhibition (96.2–100%) in both antioxidant assays, and these maximum antioxidant activities of PG are comparable to those of α-tocopherol (97.8–100%). Although PG had high maximum inhibitions, its IC50 values at 132.24 and 98.03 µg/mL, in the DPPH and ABTS assays, are clearly higher than those of α-tocopherol (25.61 and 13.56 µg/mL), so PG has moderate antioxidant activity. The DPPH radical scavenging effects of PG have been reported in numerous studies [22,23,24,26,35,36]. However, the ABTS assay results were rarely reported up until now [22]. Thus, the antioxidant data of this study also contribute to the confirmation of PG as an active molecule and enrich the available data on the ABTS radical scavenging effects of PG. 

α-Glucosidase inhibitors (aGIs) have been suggested for effective management or therapy of type 2 diabetes, a remaining serious global health issue [32,37]. Though several aGIs, such as acarbose, voglibose, and miglitol have been available, the use of these in commercial drugs has resulted in side effects [37], so the search for new natural aGI compounds is still necessary. PG and acarbose (a commercial aGI) were tested for their inhibition activities against α-glucosidase at various concentrations, and the results are presented in Figure 6. PG showed positive inhibition of α-glucosidase with the maximum inhibition level of 99% at the low concentration of 50 µg/mL, while acarbose showed a maximum inhibition of 77% at the much higher concentration of 800 µg/mL. For further comparison, the α-glucosidase inhibition activity of these two compounds was also expressed as IC50 values of 0.0183 and 328.4 µg/mL, respectively. Thus, PG was confirmed as a potent microbial aGI with higher activity compared to acarbose. PG was reported to show antidiabetic effects via anti-insulitis [38], but its novel potential as anti-α-glucosidase related to anti-type 2 diabetes activity is reported for the first time in this work. 

### 2.5. Docking Study of Prodigiosin towards an Enzyme Targetting Anti-Type 2 Diabetes 

To understand the interactions of PG (ligand) with the enzyme α-glucosidase (the target protein), a docking study was performed. Acarbose was also used for comparison. The molecular structures of ligands and target protein were prepared using MOE-2015.10 software before docking simulation. Since the in vitro assay of α-glucosidase inhibition was tested at pH 7, a virtual pH of 7 was set to prepare the structures of inhibitors and enzyme molecules. As presented in Figure 7, PG exists in two forms of neutral-PG (Figure 7a), and cation-PG (Figure 7b) in 2.7% and 97.3% proportions, while acarbose exists in neutral form (neutral-AC, Figure 7c), and all three of these ligands were subjected to a docking study with the target enzyme α-glucosidase (Figure 7d). The docking study results are presented in Table 4 and Figure 8. 

In the docking study, Root Mean Square Deviation (RMSD) was considered an important indicator to define the interactions between ligands (inhibitors) and a binding site on the target protein (enzyme); specifically, it was established whether such binding is acceptable or not. When this value reaches more than 3.0 Å, it indicates the interaction is not significant, and thus, the inhibitor displays a very weak inhibition against the target enzyme based on this binding alternative [26,39]. As shown in Table 4, all three ligands—neutral-PG, cation-PG, and neutral-AC—interacted with binding sites on the target protein with low RMSD values of 1.07, 1.13, and 1.50 Å; they showed good and acceptable interactions by binding to α-glucosidase [26,39]. In the virtual study, to identify potent inhibitors or compare efficacies among inhibitors, docking scores (DS) were used. When an inhibitor interacts with the target enzyme with a DS below −3.2 kcal/mol, it may be a potent inhibitor with good binding ability to the target enzyme [26,40]. According to the results in Table 4, all these ligands (neutral-PG, cation-PG, and neutral-AC) were potent inhibitors due to their very low DS values of −9.5, −14.6, and −10.5 kcal/mol, respectively. Of these, the neutral-PG showed a slightly lesser inhibitory effect than the neutral-AC, but the cation form of PG (cation-PG) was a much stronger inhibitor than acarbose based on their DS values. In addition, the cation-PG is recognized as the major form of PG at 97.3% based on the output from the MOE-2015.10 software. This virtual result is in agreement with the experimental in vitro tests (Figure 6).

The interactions and binding of these ligands at binding sites (BS) on the α-glucosidase enzyme are presented in Figure 8. In enzyme docking simulations, a candidate inhibitor may interact by binding on the enzyme protein at various BSs, but only one BS with the lowest binding energy was used for the detailed description of the interaction [26,41]. Based on the output from MOE-2015.10, the chosen BS on α-glucosidase was found to contain 21 amino acids, including ASP232, ILE233, ALA234, PHE236, TYR243, TRP329, ASP357, ILE358, ILE396, TRP432, TRP467, ASP469, MET470, PHE476, ARG552, TRP565, GLY567, ASP568, PHE601, ARG624, and HIS626.

As presented in Figure 8b, among the two forms of PG, neutral-PG was bound with α-glucosidase via its interactions with the two amino acids ASP 232 and LYS 506. Two linkages, including 1 pi-H and 1 pi-cation, with the recorded distances of 4.08 and 4.19 Å, respectively, were found in the interactions with ASP 232 and LYS 506, with the same 5-ring of neutral-PG. The energies (E) of these pi-H and pi-cation linkages were also recorded as −0.6 and −5.4 kcal/mol, respectively. Cation-PG was also found to bind to BS on the target enzyme via interactions with two amino acids, ASP 568 and PHE 601, but six linkages were created. Of these, amino acid ASP 568 created up to five linkages via interaction with N 2 and N 3 of cation-PG to form 3 H-donor and 2 ionic linkages, with the recorded distance and great E values in the ranges of 2.23 to 3.23 Å and −2.8 to −5.8 kcal/mol, respectively. The amino acid PHE 601 interacted with the 5-ring of cation-PG to create a pi–pi linkage with its distance and E value recorded at 3.23 Å and −0 kcal/mol, respectively. In brief, the cation form of PG had more effective interactions with the target enzyme and all of the linkages formed had very low binding energies, resulting in very low DS for cation-PG when interacting with the protein enzyme compared to the neutral form of PG.

The interaction and binding of the commercial inhibitor (neutral acarbose) are presented in Figure 8d. This ligand was found to bind at a BS on α-glucosidase via interacting with various amino acids, including ASP 568, MET 470, ASP 357, and ALA 234 to create up to nine linkages (eight H-donor, one H-acceptor) with the distances and E values recorded in the ranges of 2.65 to 3.91 Å and −0.7 to −3.3 kcal/mol, respectively. Although neutral-AC displayed more interactions than cation-PG with the target enzyme, all its linkages had larger E-values than those of cation-PG, so neutral acarbose was found to be a weaker inhibitor based on the docking simulations. As mentioned above, this virtual evaluation matched the experimental data. 

Due to its vast array of potential biological effects, the studies on PG have received much attention and have quickly increased in number in recent years [8,9,10,11,12,22,23,24,25,26]. Virtual studies on the evaluation and elucidation of the bioactivities of PG were reported earlier [26,42,43,44,45,46]. However, the enzyme inhibition for anti-diabetes activity is reported for the first time in this study, and the interactions and binding of PG with α-glucosidase are, for the first time, presented in the current work. According to the in vitro tests and the virtual evaluation, PG is a potential candidate for type 2 diabetes management. However, further investigations in various animal models as well as clinical studies are needed in the development of this potential active ingredient into anti-diabetes drugs.

## 3. Materials and Methods

### 3.1. Materials

Cassava wastewater was collected from Buon Ma Thuot, Dak Lak, Vietnam. The *S*. *marcescens* strains including *S*. *marcescens* TKU011, TNU01, TNU02, and CC17 were retained from our earlier studies [25,47,48]. Silicagel (Geduran^®^ Si 60, size: 0.040–0.063 mm) was purchased from Merck Sigma Chemical Co. (St. Louis City, MO, USA). Enzyme *Saccharomyces cerevisiae* α-glucosidase and substrate *p*-nitrophenyl glucopyranoside (*p*NPG) were acquired from Sigma Chemical Co., St. Louis City, MO, USA and Sigma Aldrich, 3050 Spruce Street, St. Louis, MO, USA, respectively. 

### 3.2. Methods

#### 3.2.1. Experiments of Prodigiosin Production via S. marcescens Fermentation

*Experiments for screening the suitable PG-producing strain:* the 4 strains of *S*. *marcescens* (TUK011, TNU01, TNU02, and CC17) were used for fermentation of cassava-based industrial wastes to produce PG. Cassava residues (CR) and cassava wastewater (CW) were supplemented with free protein (0.5% casein), 0.02% K_2_SO_4_, 0.025% Ca_3_(PO_4_)_2_, at an initial pH of 7.0, and used for fermentation by 4 strains of *S*. *Marcescens* at 28 °C for 2 days (these fermentation conditions are denoted by *). The most active strain was chosen for further tests.*The effect of free protein added into the culture medium:* casein at several concentrations (0, 0.125, 0.25, 0.5 and 0.75%) was added into the liquid culture medium containing 0.02% K_2_SO_4_, 0.025% Ca_3_(PO_4_)_2_, with an initial pH of 7.0. These designed media were fermented with the most active strain under the fermentation conditions presented above (*). It was found that 0.25% casein was the best concentration to enhance PG biosynthesis, and this was used in further investigations.*Production of PG in different commercial media:* various commercial materials, including Nutrient broth (NB), Luria-Bertani broth (LB), King’s B, soybean casein digest medium (SCDM), and casein, were used as the C/N sources for fermentation, in the same conditions (*), with *S. marcescens* TNU01. All the commercial media were used at 1% for the preparation of the culture medium. Cassava wastewater was supplemented with 0.25% casein and also used for fermentation in the above conditions (*) to compare the PG yields among these substrates.*The effects of sulfate salts added to media on PG production:* various sulfate salts (K_2_SO_4_, (NH_4_)_2_SO_4_, CaSO_4_, MgSO_4_, ZnSO_4_, and FeSO_4_) were added to culture media to explore their effects on PG production with *S*. *Marcescens* TNU01. The medium contained 0.25% casein, 0.05% sulfate salt, 0.03% K_2_HPO_4_ in cassava wastewater with an initial pH of 6.65 and MgSO_4_ was determined as the most suitable salt for PG production with the highest yield. It was further investigated in terms of the effects of its concentration (0–0.15%) in the medium on PG biosynthesis. The fermentation was performed according to the protocol (*) described above.*The effects of**phosphate salts added to media on PG production:* some phosphate salts (KH_2_PO_4_, Ca_3_(PO_4_)_2_**,** Na_2_HPO_4_, K_2_HPO_4_, and NaH_2_PO_4_) were added into culture media to explore their effects on PG production with *S*. *Marcescens* TNU01. The medium contained 0.25% casein, 0.05% MgSO_4_, 0.03% phosphate salt in cassava wastewater with an initial pH of 6.65 and K_2_HPO_4_ was the most suitable salt for PG production with the highest yield. It was further investigated for effects of its concentration (0–0.15%) in the medium on PG biosynthesis. The fermentation was performed according to the protocol (*) above.

#### 3.2.2. Experiments on Scaling-up of PG Production to 14 L-Bioreactor System

*S*. *Marcescens* TNU01 seeds were pre-cultivated in several 500 mL flasks at 28 °C for 1.5 days. The bacterial seeds (700 mL) were injected into the bioreactor system containing 6.3 L of newly mixed liquid medium including 0.25% casein, 0.05% MgSO_4_, 0.1% K_2_HPO_4_ in cassava wastewater with an initial pH of 6.65. The fermentation was performed at 28 °C in the dark for 14 h, and the PG yield was tested every 2 h. 

#### 3.2.3. Experiments for Qualification and Purification of PG

The PG qualification was performed following the protocols described in our previous work [26]. A quantity of 2.0 mL methanol was mixed with 0.25 mL of 2.0% AlK(SO_4_)_2_·12H_2_O, and 0.5 mL of fermented medium and centrifuged at 1400 × *g* for 5 min. Then 0.5 mL of collected supernatant was mixed with 4.5 mL acidic methanol adjusted by 0.5 N HCl and this mixture was measured for its optical density at 535 nm (OD_535 nm_). The PG purified in our previous work [26] was used as the standard to calculate PG yield. In this study, PG was also extracted and isolated according to the method detailed in a prior report [26]. The red pigment compound purified in this study was also analyzed by HPLC fingerprinting to confirm it as prodigiosin and assess its purity grade.

#### 3.2.4. High-Performance Liquid Chromatography (HPLC)

PG was dissolved in methanol at 1 mg/mL and 3 µL of PG (1 mg/mL) was injected into the HPLC system. The sample was separated in a C18 column using a mobile phase containing 70% MeOH in water with pH at 3.0 adjusted using 10 mM ammonium acetate, with a stable flow rate of 0.8 mL/min, and detection at 535 nm. The temperature of the column was set at 30 °C, and the analysis was conducted in 20 min.

#### 3.2.5. Bioactivity Assays

The in-vitro antidiabetic effects of PG were tested via α-glucosidase inhibitory activity. This assay was conducted according to the method earlier described by Nguyen et al. (2018) [49]. PG was prepared in DMSO to different concentrations while enzyme α-glucosidase and substrate *p*-NPG were prepared in 0.1 mol/L potassium phosphate buffer (pH 7). The assay included some typical steps [49,50], as follows:*Preincubation*: The experiments were performed in 96-well templates. A quantity of 50 μL PG solution was injected into a well containing 100 μL α-glucosidase solution, and then the mixture solution was preincubated at 37 °C for 20 min to allow the inhibitor to interact with the target enzyme.*Reaction period*: 50 μL *p*-NPG (10 mmol/L) was injected into the well containing a mixture of 100 μL α-glucosidase and 50 μL PG to start the reaction. This step maintained 37 °C for 30 min and was stopped by adding 100 μL Na_2_CO_3_ (1 mol/L).*Harvesting data and calculation*: The final mixture given above was measured at 410 nm (namely **E**). The control group was also prepared as described above, in 2 steps, but with 50 μL 0.1 mol/L potassium phosphate buffer (pH 7) instead of the PG solution, and its absorbance at 410 nm was also recorded (namely **C**). The inhibitory activity (%) was calculated as follows:
(1)$$\alpha \text{-Glucosidase inhibitory activity}\;(\%)=\frac{\left(C-E\right)}{C\times 100}$$

The IC50 value was defined as the concentration of inhibitor that inhibits 50% of enzyme activity [50]. 

Some other bioactivities, including anti-NO and antioxidant activities, were also tested. Of these, the anti-NO assay was performed following the protocol presented in our previous work, and Homogentisic acid was used as a reference compound for comparison. The antioxidant activity was evaluated using the DPPH and ABTS radical scavenging assays, as presented in prior studies [33,36], respectively, and α-tocopherol was the commercial antioxidant compound included for comparison. 

#### 3.2.6. Docking Study Performance 

The docking simulations were conducted following the protocol described in our earlier work [26]. The structural data of the target protein (α-glucosidase) was obtained from the Worldwide Protein Data Bank and its 3-D protonation was prepared using MOE QuickPrep based on the positions of the ligand within 4.5 Å and the presence of important amino acids. The active zone of the ligands on the target enzyme was found using the site finder in MOE, and all of the water molecules were removed before the recreation of enzymic action zones. ChemBioOffice 2018 software was used for the preparation of PG structures (ligands), which were further optimized using the MOE system with parameters of Force field MMFF94x; R-Field 1: 80; cutoff, Rigid water molecules, space group p1, cell size 10, 10, 10; cell shape 90, 90, 90; and gradient 0.01 RMS kcal.mol^−1^A^−2^. The docking was performed on the ligands (cation-PG, neutral-PG, neutral-acarbose) towards the enzyme α-glucosidase using MOE-2015.10 software. The data outputs, including Root Mean Square Deviation (RMSD), docking score (DS), amino acid compositions, interaction types (linkages), and interaction between amino acids with the binding site in the enzyme, and the distances of the linkages, were recorded.

### 3.3. Statistical Analysis

The experimental data were analyzed via the simple variance (ANOVA), then Fisher’s LSD test (when the experiment contained ≤ 5 items that needed to be compared) and Duncan′s multiple range test (when the experiment contained ≥ 6 items that needed to be compared) were evaluated at *p* = 0.01. Statistical Analysis Software (SAS-9.4) purchased from SAS Institute Taiwan Ltd. (Taipei, Taiwan) was used for statistical analysis.

## 4. Conclusions

This study was successful in approaching the reuse of cassava industrial wastes for the scaled-up production of bioactive PG, with regard to the cost-effectiveness of production and the consideration of environmental issues. PG was produced by *S. marcescens* TNU01 at a high-level yield of 6150 mg/L and at a large scale (7 L culture medium) in a 14 L-bioreactor system, using a novel medium that mainly contained cassava wastewater but was also supplemented with 0.25% casein, 0.05% MgSO_4_, and 0.1% K_2_HPO_4_, with fermentation at 28 °C for 8 h. The red pigment compound produced in this study was purified and evaluated for some pharmacological activities. In the experimental in vitro tests, the purified PG displayed moderate antioxidant, and high anti-NO activities. Notably, PG was found to be a novel candidate α-glucosidase inhibitor. In both in vitro and docking study evaluations, PG demonstrated stronger effective inhibition of α-glucosidase and better binding energy than acarbose, a commercial antidiabetic drug. The results from this work suggest that cassava wastewater is a low-cost and environmentally friendly material for the production of PG via fermentation, and PG is a novel candidate active ingredient for drug development targeting antidiabetic management. 

## Figures and Tables

**Figure 1 molecules-26-06270-f001:**
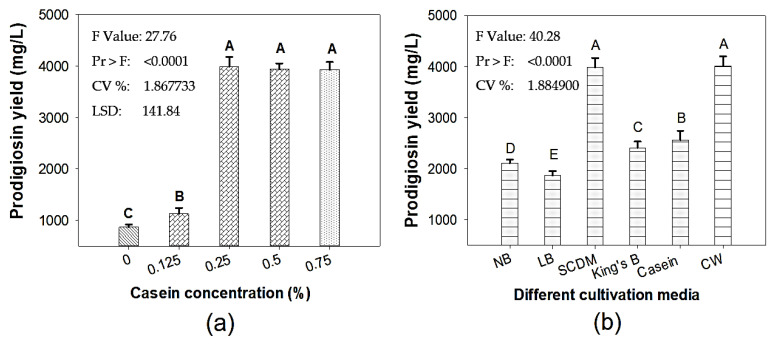
The effects of free protein added to the culture medium (**a**), and the effects of different cultivation media (**b**) on PG production with *S*. *marcescens* TNU01 fermentation. Standard errors (SD) are shown as error bars. The data were analyzed via the simple variance (ANOVA), then Fisher’s LSD test (**a**), and Duncan′s multiple range test (**b**) were evaluated at *p* = 0.01. Means of prodigiosin yield values with the same letter in each figure are not significantly different.

**Figure 2 molecules-26-06270-f002:**
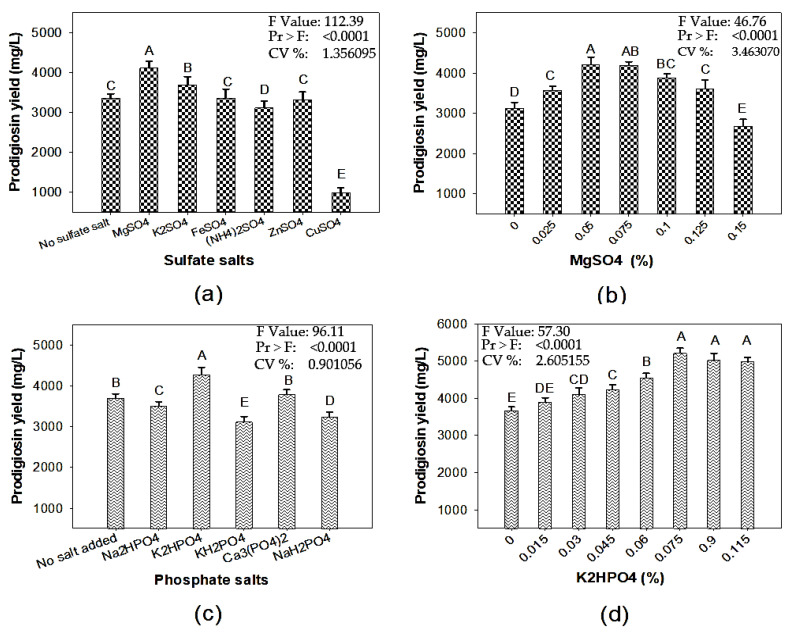
The effects of different kinds of sulfate salts (**a**), supplementary MgSO_4_ concentration (**b**), different kinds of phosphate salts (**c**), and supplementary K_2_HPO_4_ concentration (**d**) on prodigiosin biosynthesis using *S*. *marcescens* TNU01 from a medium based on cassava wastewater. All experiments were conducted in triplicate. The data were analyzed via the simple variance (ANOVA), then Duncan′s multiple range test was conducted at *p* = 0.01. Means of prodigiosin yield values with the same letter in each figure are not significantly different. Standard errors (SD) are shown as error bars.

**Figure 3 molecules-26-06270-f003:**
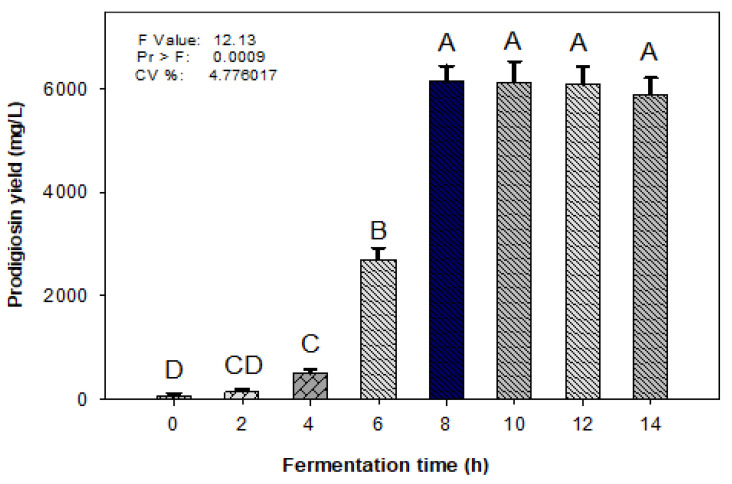
Bioconversion of medium based on cassava waste to prodigiosin in a 14 L-bioreactor. All experiments were conducted in triplicate. The data were analyzed via the simple variance (ANOVA), then Duncan′s multiple range test was performed at *p* = 0.01. Means of prodigiosin yield values with the same letter in each figure are not significantly different. Standard errors (SD) are shown as error bars.

**Figure 4 molecules-26-06270-f004:**
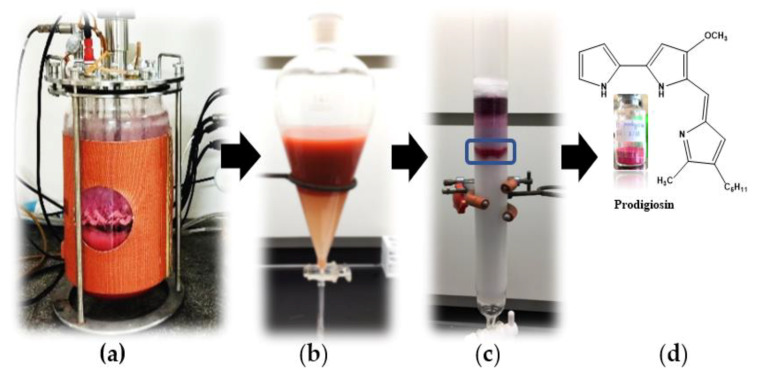
Extraction and purification of bioactive prodigiosin (PG) from fermentation with *S. marcescens* TNU01. The liquid medium, containing cassava wastewater fermented with *S. marcescens* TNU01 in a 14 L-bioreactor system (**a**), was subjected to layer separation using ethyl acetate (**b**), and the crude PG was further isolated in an open silica column (**c**) to obtain purified PG (**d**).

**Figure 5 molecules-26-06270-f005:**
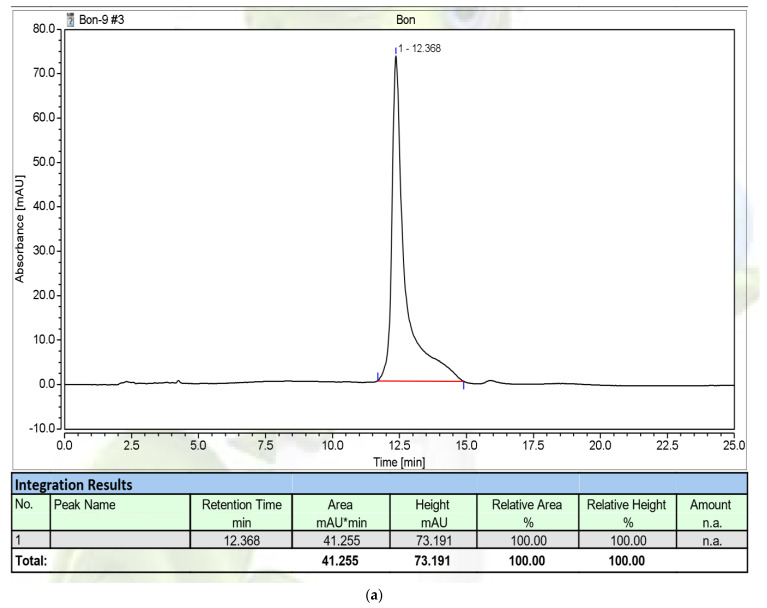
The high-performance liquid chromatography (HPLC) profiles of prodigiosin purified in this work (**a**), and a reference PG compound (**b**) obtained in our previous study [26]. The retention time (RT) of purified prodigiosin and the reference PG was recorded at 12.368 min and 12.425 min, respectively.

**Figure 6 molecules-26-06270-f006:**
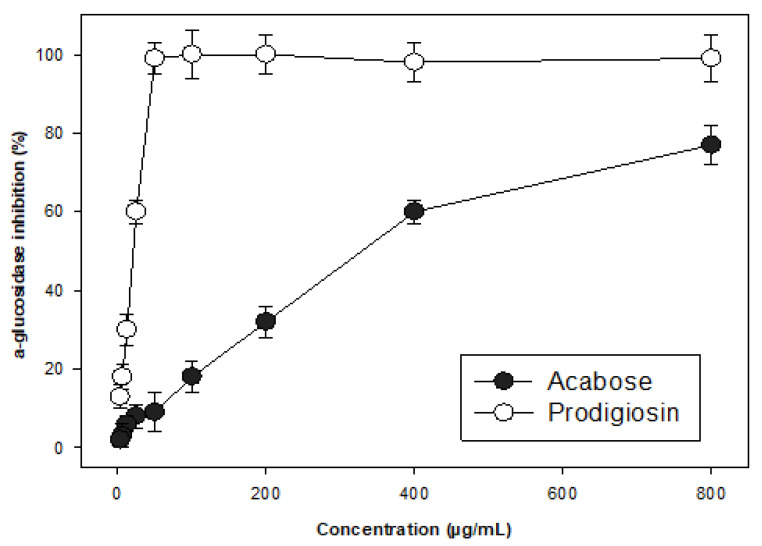
The α-glucosidase inhibitory activities of prodigiosin and acarbose.

**Figure 7 molecules-26-06270-f007:**
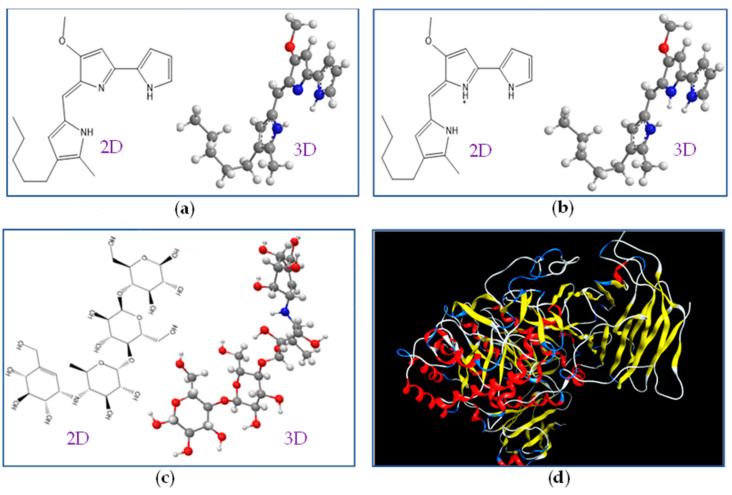
The 2-D and 3-D structures of neutral-PG (**a**), cation-PG (**b**), neutral-AC (**c**), and the α-glucosidase enzyme’s crystal structure (**d**).

**Figure 8 molecules-26-06270-f008:**
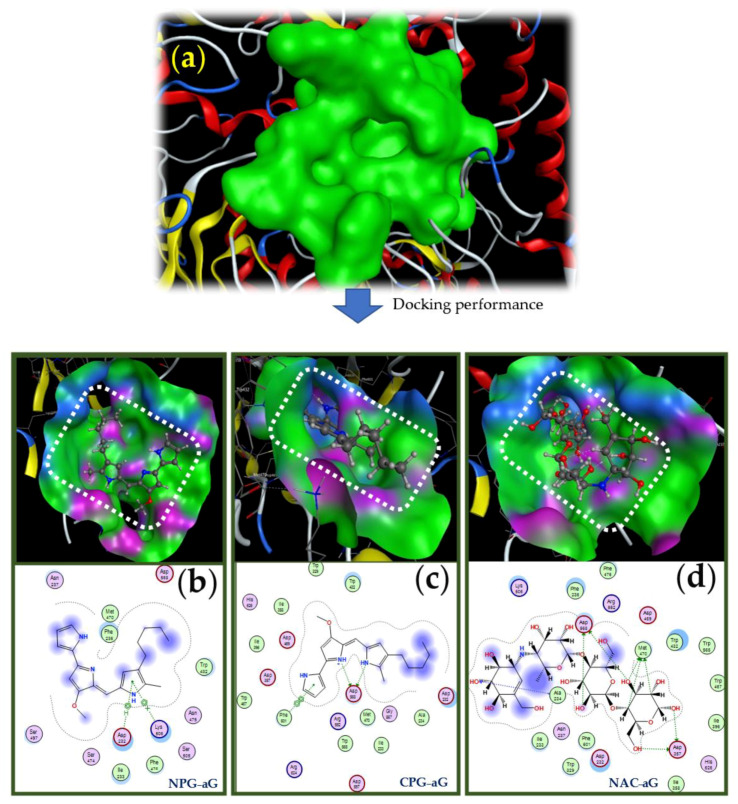
The docking study of prodigiosin and acarbose with α-glucosidase enzyme. The 3-D structure of the binding site on α-glucosidase (**a**). The interactions and binding of the ligands neutral-PG, cation-PG, and neutral-AC with the α-glucosidase enzyme are labeled as NPG-aG (**b**), CPG-aG (**c**), and NAC-aG (**d**).

**Table 1 molecules-26-06270-t001:** Experiments for screening the most suitable PG-producing strain.

PG-Producing Bacterial Strain	PG Concentration (mg/L)	
	CR	CW
*S*. *marcescens* TKU011	1774 ± 33 ^G^	3730 ± 54 ^C^
*S*. *marcescens* TNU01	2191 ± 41 ^E^	3981 ± 32 ^A^
*S*. *marcescens* TNU02	1921 ± 51 ^F^	3811 ± 101 ^B^
*S*. *marcescens* CC17	1532 ± 45 ^H^	3474 ± 211 ^D^
Control	PG not detected	PG not detected
F Value	35.41	
Pr > F	<0.0001	
CV %	0.835473	

All experiments were conducted in triplicate. The data were analyzed via the simple variance (ANOVA), then Duncan′s multiple range test was conducted at *p* = 0.01. Values in the same column with different letters are significantly different.

**Table 2 molecules-26-06270-t002:** Prodigiosin production, at a large scale, reported by various studies.

PG-Producing Strain	Major C/N Sources	Reactor Size (L)	PG Yield (mg/L)	Fermentation Time (h)	Reference
*S. marcescens* TNU01	Cassava wastewater/0.25% casein	7.0	6150	8	This report
*S. marcescens* CC17	1.35% SHP/0.15% casein	6.75	6310	8	[26]
*S. marcescens* TNU01	1.12% deSSP/0.48% casein	5.0	6200	8	[22]
*S. marcescens* TNU02	1.12% deCSP/0.48% casein	4.5	5100	8	[23]
*S. marcescens* TNU01	1.75% SPP	3.0	3450	12	[24]
*S. marcescens* BS 303 (ATCC^®^ 13880™)	3.0% glycerol/1.05% casein peptone	0.935	872	65	[28]
*S. marcescens*	0.865% sucrose/0.662% peptone	6.5	595	52	[29]
*S. marcescens* 02	1.0% glycerol, 1.0% tryptone, 1.0% extract of yeast	2.75	583	20	[30]
*Chryseobacterium artocarpi* CECT 849	1.125% Lactose and 0.6% l-tryptophan.	50	522	24	[31]

SHP—shrimp head powder; deSSP—demineralized shrimp shell powder; deCSP—demineralized crab shell powder; SPP—squid pen powder.

**Table 3 molecules-26-06270-t003:** The anti-NO and antioxidant effects of prodigiosin.

Activity	Compounds	Max Inhibition (%, Tested Concentration)	IC50 (µg/mL)
Anti-NO activity	Prodigiosin	91.3% (80 µg/mL)	18.06
	Homogentisic acid	92.1% (80 µg/mL)	16.12
DPPH assay (Antioxidant activity)	Prodigiosin	100% (380 µg/mL)	132.34
	α-tocopherol	100% (50 µg/mL)	25.61
ABTS assay (Antioxidant activity)	Prodigiosin	96.2% (2 mg/mL)	98.03
	α-tocopherol	97.8% (50 µg/mL)	13.56

**Table 4 molecules-26-06270-t004:** The docking simulation results of ligand binding with target enzyme α-glucosidase.

Ligand Form (Inhibitor Candidate)	Symbol (Inhibitor-Enzyme)	RMSD (Å)	DS (kcal/mol)	Number of Interactions	Amino Acids Interacting with the Ligand (Distance (Å)/E (kcal/mol)/Linkage Type)
Neutral-PG	NPG–aG	1.07	−9.5	2 linkages (1 pi-H, 1 pi-cation)	ASP 232 (4.08/−0.6/pi-H) LYS 506 (4.19/−5.4/pi-cation)
Cation-PG	CPG–aG	1.13	−14.6	6 linkages (3 H-donor, 2 ionic, and 1 pi–pi)	ASP 568 (2.74/−3.7/H-donor) ASP 568 (2.83/−2.8/H-donor) ASP 568 (3.23/−4.9/H-donor) ASP 568 (2.38/−5.8/ionic) ASP 568 (3.23/−3.1/ionic) PHE 601 (3.23/−0/ pi–pi)
Neutral-AC	NAC–aG	1.50	−10.5	9 linkages (8 H-donor, 1 H-acceptor)	ASP 568 (2.81/−1.3/H-donor) MET 470 (3.76/−1.0 H-donor) ASP 568 (2.69/−3.2/H-donor) ASP 568 (2.96/−1.5/H-donor) ASP 357 (3.03/−3.3/H-donor) ASP 357 (2.65/−1.9/ H-donor) MET 470 (3.91/−0.7/H-donor) MET 470 (3.90/−0.7/H-donor) ALA 234 (3.00/−0.8/H-acceptor)

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
