# Peer review of "Utilization of Cassava Wastewater for Low-Cost Production of Prodigiosin via Serratia marcescens TNU01 Fermentation and Its Novel Potent α-Glucosidase Inhibitory Effect"

_molecules, 2021, doi:10.3390/molecules26206270_

Round 1

Reviewer 1 Report

The manuscript is interesting, it treats a good scientific issue.  The authors try to scale-up the production of Prodigiosin from cassava wastewater by Serratia marcescens. They examined the product as an antioxidant and as a novel candidate α-glucosidase inhibitor using docking study.

However, the manuscript is well designed and good written, it still needs a revision before decision of the editor.

The following are some comments that should be treated carefully.

  • Do the authors carry out their experiments in 7-L or 14-L bioreactor? The two types were mentioned may times in through the text, please clarify this point.
  • I suggest mentioning the producing bacterium in the title of the manuscript!
  • In abstract you should spell out the name of the bacteria because you mention it for the first time.
  • Please revise the abstract carefully for some misstops, for example:
  • 0183μg/mL, 70μg/ml (add a space and use uniform units) and revise in the same manure the manuscript thoroughly.
  • In keyword: change Marcescens to S. marcescens
  • In conclusions: change K2HPO4 to K2HPO4
  • I emphasis on the revision of whole manuscript for such above-mentioned mistakes.

Author Response

Dear Reviewer/Advancer

We feel pleasure to thank you for your time and effort, as well as your excellent suggestions for refining the readability and impact of the manuscript. We have gone through all the suggestions cautiously and made the revisions accordingly; and all amended parts have been typed in red in the revised manuscript. Finally, we like to express our deep thanks to your comments and suggestions again. You certainly have served to improve the quality of this paper. We hope our response is satisfactory.

Looking forward to hearing from you.               

Thanking you,

Yours Sincerely,

Dr. Van Bon Nguyen

Comments and Suggestions for Authors:

The manuscript is interesting, it treats a good scientific issue.  The authors try to scale-up the production of Prodigiosin from cassava wastewater by Serratia marcescens. They examined the product as an antioxidant and as a novel candidate α-glucosidase inhibitor using docking study.

However, the manuscript is well designed and good written, it still needs a revision before decision of the editor.

The following are some comments that should be treated carefully.

Reply: Thank you for giving us very positive comments and valuable revisions.  

Do the authors carry out their experiments in 7-L or 14-L bioreactor? The two types were mentioned may times in through the text, please clarify this point.

Reply: The experiments of scale-up production of prodigiosin were in 14 L - bioreactor. Seven liter of cuture medium was fermented in this bioreactor. This iterm was revised according to the comments for avoiding misunderstand. Thank you for your reminding.   

I suggest mentioning the producing bacterium in the title of the manuscript!

Reply: The producing bacterium was mentioned in the tittle. Thank you.

In abstract you should spell out the name of the bacteria because you mention it for the first time.

Please revise the abstract carefully for some misstops, for example:

0183μg/mL, 70μg/ml (add a space and use uniform units) and revise in the same manure the manuscript thoroughly.

Reply: Thank you for your revision. All these items were revised.

In keyword: change Marcescens to S. marcescens

Reply: S. Marcescens was changed to S. marcescens.

In conclusions: change K2HPO4 to K2HPO4

Reply: K2HPO4 was changed to K2HPO4.

I emphasis on the revision of whole manuscript for such above-mentioned mistakes.

Reply: We already revised accordingly whole manuscript. One more time, we would like to pay our deeply thanks for your very careful review, and valuable comments for enhancement of the quality of the manuscript.

Reviewer 2 Report

This study investigated the potential of cassava-based industrial wastes as substrate for prodigiosin (PG) production by S. marcescens and its potential application as anti-diabetic drug, based on in vitro and docking studies. Cassava wastewater was tried as a cost-effective substrate for PG production.  The manuscript is properly written. For PG production, the fermentation conditions were optimized using one factor at a time. The production has been scaled up in 14 L bioreactor. The purified PG showed moderate antioxidant and high anti-NO activities. In addition, PG was found as a α-glucosidase inhibitor in both in vitro and docking study evaluations. The following modifications should be amended in the manuscript before considering for publication.

Line 25: In the abstract give the full genus name of S. marcescens

Fig 1b should be cited in line #102

Authors should represent the PG productivity in mg/L (to maintain the uniformity) for all the experimental results including figures. When reading it gives an impression that PG productivity increased from 5.202 to 6150. However, the first one is mg/mL and the next one is mg/L.

Do the statistical analysis for the results and represent the significant differences in the figure. To me, in Fig. 3, it looks like 10 h is giving better yield than 8 h. The statistical analysis will eliminate this misconception. I recommend doing ANOVA and a post hoc analysis.

Give the full form of anti-NO when you used for the first time.

Author Response

Detailed response to reviewers' comments

 Manuscript ID: molecules-1411092

The Original Title: “Utilization of Cassava Wastewater for Low-cost Production of Prodigiosin via Fermentation Technology and Its Novel Potent α-Glucosidase Inhibitory Effect

 Dear Reviewer/Advancer

We feel pleasure to thank you for your time and effort, as well as your excellent suggestions for refining the readability and impact of the manuscript. We have gone through all the suggestions cautiously and made the revisions accordingly; and all amended parts have been typed in red in the revised manuscript. Finally, we like to express our deep thanks to your comments and suggestions again. You certainly have served to improve the quality of this paper. We hope our response is satisfactory.

Looking forward to hearing from you.               

Thanking you,

 Yours Sincerely,

Dr. Van Bon Nguyen

Comments and Suggestions for Authors:

This study investigated the potential of cassava-based industrial wastes as substrate for prodigiosin (PG) production by S. marcescens and its potential application as anti-diabetic drug, based on in vitro and docking studies. Cassava wastewater was tried as a cost-effective substrate for PG production.  The manuscript is properly written. For PG production, the fermentation conditions were optimized using one factor at a time. The production has been scaled up in 14 L bioreactor. The purified PG showed moderate antioxidant and high anti-NO activities. In addition, PG was found as a α-glucosidase inhibitor in both in vitro and docking study evaluations. The following modifications should be amended in the manuscript before considering for publication.

Reply: Thank you for giving us very positive comments and valuable revisions.  

Line 25: In the abstract give the full genus name of S. marcescens

Reply: Thank you, this item was revised according to the comment.

Fig 1b should be cited in line #102

Reply: Thank for your revision, this figure (1b) was cited corrected. 

Authors should represent the PG productivity in mg/L (to maintain the uniformity) for all the experimental results including figures. When reading it gives an impression that PG productivity increased from 5.202 to 6150. However, the first one is mg/mL and the next one is mg/L.

Reply: The unit mg/mL was changed to mg/L in all the figures and texts in whole the manuscript.

Do the statistical analysis for the results and represent the significant differences in the figure. To me, in Fig. 3, it looks like 10 h is giving better yield than 8 h. The statistical analysis will eliminate this misconception. I recommend doing ANOVA and a post hoc analysis.

Reply: Thanks for your remiding, we already make statistical analysis for the results and presented in almost the tables and figures. ANOVA and a post hoc analysis were used according to your suggestion.  Base on the Duncan′s multiple range test at p = 0.01, the yield of prodigiosin produced at 8 h and 10 h in bioreactor system are at the same level (ranked at A level).

Give the full form of anti-NO when you used for the first time.

Reply: This item was revised. One more time, we would like to pay our deeply thanks for your very careful review, and valuable comments for enhancement of the quality of the manuscript.